# All Set before Flowering: A 16S Gene Amplicon-Based Analysis of the Root Microbiome Recruited by Common Bean (*Phaseolus vulgaris*) in Its Centre of Domestication

**DOI:** 10.3390/plants11131631

**Published:** 2022-06-21

**Authors:** Francisco Medina-Paz, Luis Herrera-Estrella, Martin Heil

**Affiliations:** 1Laboratorio de Ecología de Plantas, Departamento de Ingeniería Genética, Centro de Investigación y de Estudios Avanzados (CINVESTAV)—Unidad Irapuato, Irapuato 36824, GTO, Mexico; francisco.medina@cinvestav.mx; 2Laboratorio Nacional de Genómica para la Biodiversidad, Centro de Investigación y de Estudios Avanzados (CINVESTAV)—Unidad de Genómica Avanzada, Irapuato 36824, GTO, Mexico; lherrerae@cinvestav.mx or; 3Institute of Genomics for Crop Abiotic Stress Tolerance, Texas Tech University, Lubbock, TX 79424, USA

**Keywords:** bacterial community structure, dry bean, root endosphere, Mexico, native soil, rhizobia, rhizosphere, root microbiome

## Abstract

Plant roots recruit most prokaryotic members of their root microbiota from the locally available inoculum, but knowledge on the contribution of native microorganisms to the root microbiota of crops in native versus non-native areas remains scarce. We grew common bean (*Phaseolus vulgaris)* at a field site in its centre of domestication to characterise rhizosphere and endosphere bacterial communities at the vegetative, flowering, and pod filling stage. 16S r RNA gene amplicon sequencing of ten samples yielded 9,401,757 reads, of which 8,344,070 were assigned to 17,352 operational taxonomic units (OTUs). Rhizosphere communities were four times more diverse than in the endosphere and dominated by Actinobacteria, Bacteroidetes, Crenarchaeota, and Proteobacteria (endosphere: 99% Proteobacteria). We also detected high abundances of Gemmatimonadetes (6%), Chloroflexi (4%), and the archaeal phylum Thaumarchaeota (Candidatus Nitrososphaera: 11.5%): taxa less frequently reported from common bean rhizosphere. Among 154 OTUs with different abundances between vegetative and flowering stage, we detected increased read numbers of *Chryseobacterium* in the endosphere and a 40-fold increase in the abundances of OTUs classified as *Rhizobium* and *Aeromonas* (equivalent to 1.5% and over 6% of all reads in the rhizosphere). Our results indicate that bean recruits specific taxa into its microbiome when growing ‘at home’.

## 1. Introduction

Plant roots harbour highly diverse microbiotas that contribute important functions to the nutrition and health of their host, similarly to the well-known beneficial effects of the human gut microbiome [1,2]. Since most prokaryotic members of the root microbiome are recruited from the surrounding bulk soil, roots secrete organic molecules and release cells from the root caps to generate a nutrient-rich zone in the directly adjacent soil, called the ‘rhizosphere’ [3,4]. Free-living soil bacteria employ positive chemotaxis to reach this zone and most colonize the rhizosphere, while few strains enter the roots to colonize the ‘root endosphere’. This horizontal mode of transmission represents a striking difference from the human gut and opens the question to which degree plants can control the composition of their microbiota. The endosphere represents an environment that is mainly controlled by the plant and its colonization requires specific adaptations [5,6,7,8], but plants have less control over the rhizosphere. In fact, soil type and compartment (i.e., rhizosphere versus endosphere) are usually identified as the main determinators of the composition of root-associated microbiota, while plant genotype has lower or even no detectable effects [1,4,9,10,11]. Nevertheless, colonizing the rhizosphere and reaching high abundances in this zone requires specific functions and most of them intrinsically generate benefits for the host plant: examples include nutrient mobilization, the decomposition of organic macromolecules, and direct antibiosis [1,2,4,12,13,14]. Therefore, most microbiotas comprise a core set of ubiquitous taxa that usually occur at high abundances—the so-called ‘core microbiomes’ [15,16,17] that are candidates to be used as plant growth-promoting rhizobacteria (PGPR) or ‘biocontrol’ agents for future crop management strategies [1,2,4,12,13,14]. Successful application of these consortia requires knowledge of their taxonomic composition and moreover, a detailed understanding of the mechanisms that underlie their assembly. Reaching these goals has been identified as ‘research priorities’ [15]. The assembly of a microbiome is driven by complex interactions between the host plant and its microbiome as well as by interactions among the various microorganisms, frequently generating non-linear processes that are further shaped by environmental factors. Evidently, this complexity hardly allows transferring knowledge from models obtained from research using *Arabidopsis thaliana* to crops, or from one crop species to another. Therefore, a successful application of microbiomes will depend on a significant diversification of model systems [15].

For example, the plant genotype is frequently reported to have little – or no detectable - influence on the formation of root microbiomes (at least, in terms of statistically significant effects of the plant genotype on rhizosphere microbiomes that form under standardised laboratory conditions). Nevertheless, root microbiota of wild and domesticated genotypes of several crop species have been shown to differ when growing in the same soil [18] (examples comprise sugar beet [19], sunflower [20], barley [10], maize [21,22], rice [23], and bean [24]). Moreover, studies using plants that grow under controlled conditions in a standardised soil are prone to miss non-ubiquitous microbial taxa. [25,26]. Plants and their microbiota are increasingly being understood as co-evolving systems [27], but most crops are cultivated predominantly outside of their native habitat range (ex-situ), i.e., in the absence of the native microbial taxa with which they co-evolved. Thus, crops might have lost important members of their ‘original’ microbiome [28]. In fact, specific native strains were absent from the microbiota of ex-situ cultivated *Agave* spp., canola, squash and maize [21,29,30,31], but also for non-crops including Arabidopsis and diverse invasive wild species [32,33,34] (for reviews see [34,35,36]).

Although—as expressed by Stopnisek and Shade [37]—“taxa not consistently detected in the rhizosphere of a particular species, by deduction, could not be of universal importance for the host plant”, non-universal components of plant-associated microbiota—including those that establish ‘heritable plant–microbe interactions’—have the potential to improve the net benefits for the host plant [38]. In particular, the so-called ‘hub taxa’ [39] can significantly shape the further assembly of the microbial community [39]. For example, the colonization of bean roots by *Chryseobacterium balustinum* triggered changes in the secretion of flavonoids that subsequently affected the expression of nod genes in several *Rhizobium* strains [40]. Similarly, changing abundances of *Bacillus* spp. in the rhizosphere of soybean plants grown in soils from different habitats subsequently caused differences in the composition of nodule-forming symbiotic rhizobial communities and thus, in the nitrogen-fixing activity of nodules [41].

In summary, available evidence increasingly supports (i) that ‘native’ members of the microbiome might have been lost during the ex-situ cultivation of modern crop varieties, (ii) that hub species are not necessarily identical with the members of the core microbiota [37], (iii) that native (and other non-universal) taxa have the potential to improve—or complement—the essential functions provided by the core microbiome, and (iv) that rhizobacteria which are responsive to differences in among genotypes remain to be ascertained for agronomically important crops [26]. Altogether, these findings call for mining the phytobiomes of plants grown in situ within their native range for candidates of beneficial microbial associations [18,28,35]. 

The central aim of our present study was to make a first step in this direction, using a widely cultivated but understudied crop. Therefore, we characterised the bacterial root microbiota of common bean (*Phaseolus vulgaris*) plants growing in situ in field soil within the native range and area of domestication of this crop. Common bean (or dry bean) is considered the most important food legume for direct human consumption in the world because it represents the dominant source of protein for a considerable part of the population, particularly in most countries in Latin America, Africa, the Middle East, and several Asian countries [42]. The native range of wild *P. vulgaris* is restricted to Central and South America, and all available evidence supports two independent domestication centres, which gave rise to a Mesoamerican and an Andean genetic pool of cultivated common bean [43,44,45,46,47,48]. In few words, common bean is mainly cultivated outside its native range and outside of the centres of domestication, a characteristic that common bean shares with most important crops. More importantly though, bean remains a strongly understudied crop in spite of calls to rank *P. vulgaris* as a ‘model legume’ [49]. In particular, the root-associated microbiotas of bean have received little attention, evidently except the long history of studies focused on the symbiosis with nodulating rhizobia [50,51]. Relatively few culture-dependent screenings aimed to identify and characterise growth-promoting or disease suppressing microbial strains [52,53,54,55,56,57,58] and an even lower number of studies used culture-independent methods to explore the composition and putative importance of the prokaryotic root microbiome of bean (see Table 1 for a compilation). A pioneering study used culture-dependent methods to isolate vertically transmitted bacterial endosymbionts from seeds and from the roots of seedlings kept under sterile conditions for [53], an effort recently complemented by a study that cultured bacteria from rhizosphere and endosphere samples of field-grown plants [52]. Three amplicon-based studies used plants grown under greenhouse conditions to compare the bacterial rhizosphere microbiota among genotypes with different domestication status (wild accessions, land races, and modern cultivars), different domestication background (Andean vs Mesoamerican genetic pools), or different levels of resistance to the fungal pathogen, *Fusarium oxysporum* [24,59,60]. One study generated genetically transformed plants to study the effect of root architecture on the endosphere microbiome [61] and most recently, two field studies explored microbial communities in the rhizosphere of different cultivars of common bean grown in various agricultural regions in the USA and Colombia, respectively [37,62], while another study explored the inter-individual and intra-individual variation among microbial communities in the seed endosphere [63] (Table 1).

**Table 1 plants-11-01631-t001:** Compilation of studies into bacterial root microbiota of common bean (*Phaseolus vulgaris*).

Year	Compartment	Conditions	Plant Stage	16S *	Size	Main Result	Ref
2010	Endosphere (seed and roots)	Sterile	Seedling 3 d	primers fD1 rD1	1500 bp	Dominating phyla: Firmicutes, Actinobacteria, α-β-and γ-Proteobacteria; genera: *Acinetobacter*, *Bacillus*, *Methylobacterium*, *Micrococcus*, *Paenibacillus*, *Rhizobium*, *Staphylococcus*	[53]
2017	Rhizosphere	Pots, GH 8 genotypes (2 WA, 3 LR, 3 MC)	Flowering	V3–V4	460 bp	Genotype explained only 13% of bact. diversity, MC harboured less Bacteroidetes than wild accessions and landraces	[59]
2018	Rhizosphere	Pots, GH, Amazon dark earth + agric. soil, 2 cultivars diff resistance to *F. oxy.*	Flowering	V3–V4 & shotgun metagenome	460 bp	More diverse and connected community in the *F. oxy*-resistant cultivar, dominated by Pseudomonadaceae, Bacillaceae, Solibacteraceae and Cytophagaceae.	[60]
2019	Rhizosphere	Pots, GH, Amazon dark earth + agric. soil 8 genotypes (2 WA, 3 LR, 3 MC)	Flowering	V3–V4	460 bp	Effects of bean genotype stronger in the agricultural soil, among a total of 15,925 OTUs, 113 highly abundant OTUs (26% of all reads) conform a core microbiome shared by all accession x soil combinations	[24]
2020	Endosphere and Rhizosphere	Open field ex-situ	Flowering	primers fD1 rD1	1500 bp	12 out of 90 cultured strains exhibited direct antibiosis: 7 *Bacillus*, 2 *Pseudomonas*, 1 *Agrobacterium,* 1 *Glutamicibacter*	[52]
2020	Endosphere and bulk soil	Pots, GH 2 genotypes w. different root morphology	Plantlet 15 d	V3		No difference between genotypes, OTU richness and diversity in soil much higher than in vermiculite	[61]
2021	Rhizosphere and bulk soil	Open field ex-situ, 2 cultivars	Vegetative Flowering Pod filling Pods ripe	V4–V5	300 bp	Weak/no effect of plant genotype, while location and soil properties as main determinants generate a biogeographic pattern of bacterial community structures	[37]
2022	Rhizosphere and bulk soil	Open field ex-situ, 2 cultivars	Vegetative	V4	290 bp	More cultivar-exclusive than shared OTUs, taxa rhizosphere of biofortified cultivar enriched in diverse groups, e.g., *Burkholderia* and *Rhodanobacter*	[62]
2022	Seed endosphere	1 cultivar	Seeds	V4	300 bp	Bacterial seed endosphere communities show inter- but not intra-individual variation	[63]
2022	Endosphere and Rhizosphere	Open field in domestication area	Vegetative Flowering Pod filling	V5–V9	750 bp		This study

* Abbreviations: 16S, primers used or regions of the 16S rRNA gene amplified; bp, base pairs; *F. oxy*, *Fusarium oxysporum*, GH, greenhouse; LR, landrace; MC, ‘modern’ cultivar’; OTU, operational taxonomic unit; WA, wild accession.

The beforementioned studies confirm the importance of soil parameters on the rhizosphere microbiota and the existence of a core set of ubiquitous—and usually abundant—taxa, but also reveal that genotype x soil interactions shape the structure of these microbiotas to an as-yet underestimated degree, as indicated by the differences observed among field sites and the enrichment of specific taxa in the rhizosphere of a biofortified cultivar (Table 1). The two studies using plants grown in agricultural fields confirm the relevance of non-universal and non-abundant strains recruited from the locally available inoculum, even when plants grow outside of their native range. We designed our present explorative study to complement the previously published work with data on the microbiome that bean plants recruit when growing in situ within the area of domestication. Since most of the earlier studies focused on the rhizosphere sampled at a single ontogenetic stage, we decided to use 16S rRNA gene amplicon sequencing for paired assessments of the rhizosphere and endosphere communities at three ontogenetic stages and focus our analyses on taxa not previously reported for bean that could represent functionally important elements of the native microbiome.

Microbiome research is driven by an ever-increasing spectrum of available methods, each one coming with its specific advantages and disadvantages. Cultivation-dependent methods remain the gold standard for any type of experimental work, but they are necessarily limited to a reduced number of culturable strains. Among the cultivation-independent methods, marker-based studies generate a—theoretically nonbiased—general snapshot of the community at affordable costs, but functional interpretations remain speculative. Shotgun metagenomic sequencing allows for taxonomic profiling and reveals the complete set of functions potentially available in a microbial community, while metatrancriptomic approaches even allow to focus on the analyses of those functions that currently are being expressed in the community. Drawbacks of the last two approaches include the considerable economic costs and the risk to miss taxa that contribute important functions to a microbiome without reaching high abundances, particularly if—as in endosphere communities—most sequencing depth is lost due to the dominance of host-derived genes. Therefore, we opted for 16S rRNA gene amplicon sequencing for this first, exploratory study and used a bioinformatic tool to predict potential gene functions of interest.

We discovered relatively large abundances of OTUs assigned to the phyla Chloroflexi and Gemmatimonadetes and of the archaeal phylum Thaumarchaeota: taxa not commonly reported from bean microbiota. Moreover, we observed that the abundances of an endosphere OTU annotated as *Chryseobacterium* and of 154 rhizosphere OTUs including *Rhizobium* and *Aeromonas* strongly increased from the vegetative to the flowering stage. These changes are consistent with a scenario of an ontogenetic shift driven by *Chryseobacterium* as a hub-taxon that facilitates the subsequent accumulation of beneficial ‘core’ taxa in the rhizosphere before the plant enters the reproductive stage. Moreover, we observed several cases in which predicted gene functions appeared enriched in a compartment that differed from the expected pattern and interpret these patterns as indicators of an as-yet underestimated importance of these functions. We consider our study as further support of the existence of microbial taxa that bean plants recruit when growing ‘at home’, which merit consideration as potentially important components of the native microbiome that have been lost during the ex-situ cultivation of this crop.

## 2. Results and Discussion

### 2.1. Validation of Experimental Design and Overall Bacterial Diversity

For the present study, we grew *Phaseolus vulgaris* (cultivar Flor de Junio Marcela) from surface-sterilized seeds sown directly into the soil in a field plot at CINVESTAV—Irapuato (1800 m above sea level; 20°43′13″ N; 101°19′43″ W), a site located within the diversity centre of wild *P. vulgaris* and the area considered to be the Mesoamerican domestication centre of bean [43,47,64] (see Section 3 for details). At each of three ontogenetic stages (early vegetative stage, flowering stage, and pod filling stage, corresponding to stages 2–4 as described in [37]), we harvested the root systems of three individual plants and collected the soil directly adhered to the roots as rhizosphere samples and—after removing nodules—the surface-sterilized roots as endosphere samples [65]. This sampling effort was duplicated for the first two stages, but not the last stage, yielding a total of ten samples (see Appendix A).

One of the major obstacles to the generation of 16S rRNA gene amplicons from different types of samples is to avoid an underestimation of the real bacterial diversity due to the use of too selective primers versus the loss of sequencing depth due to the amplification of non-target sequences from organelles [4,66,67]. For the present study, we selected the primers 799f and 1492r, a primer pair designed for the study of endophytic communities that amplifies most bacterial 16S rRNA sequences, excludes chloroplast DNA sequences, and generates an amplification product of ~750 bp from bacterial versus ~1300 bp from mitochondrial 16S rRNA genes [68]. Considering recent reports on the non-efficient amplification of bacterial 16S rRNA gene sequences from plant tissues rich in plastids [69], we used genomic DNA isolated with the hot CTAB method from sterilised bean root tissue to verify this separation of bacterial versus mitochondrial PCR products by size. Indeed, we observed two PCR bands. Cloning the two PCR bands and aligning the sequences of 28 positive clones against the NCBI nucleotide database, we confirmed that all 13 inserts obtained from the ~750 bp band corresponded to 16S rRNA genes of *Rhizobium* spp. while all 15 inserts from the ~1300 bp band corresponded to *P. vulgaris* mitochondrial DNA (Appendix A).

Therefore, we extracted the total DNA from the beforementioned rhizosphere and endosphere samples, used the primers 799f and 1492r to amplify 16S rRNA gene sequences, and subsequently—in the case of the endosphere samples—gel-electrophoresis to select the ~750 bp product. High-throughput sequencing of the amplicons on the MiSeq 2 × 300 Illumina platform yielded a total of 9,401,757 high-quality, non-chimeric reads across all samples. Read numbers obtained from the rhizosphere samples ranged from 933,482–1,127,470 (average 1,000,048) and thus were consistently higher than those from the corresponding endosphere samples (range 749,457–946,019; average 880,303) (see Appendix A for numbers of reads). Using the QIIME [70] closed reference approach against the greengenes 13_8 database at 97% sequence similarity, 8,344,070 of the reads could be assigned to a total of 17,352 operational taxonomic units (OTUs). Rarefaction curves indicate that the sequencing depth was sufficient to reach saturation for the endosphere, but not the rhizosphere samples (Figure 1A). Overall, the bacterial community in the rhizosphere was composed of a total of 16,948 OTUs (ranging from 10,089 to 11,537 per sample) and thus, was ca. 4-times richer than in the endosphere, with a total of 4176 OTUs (ranging from 783 to 2579 OTUs per sample) (Figure 1A). Shannon’s diversity indices (*H’*) confirmed that the prokaryotic communities in the rhizosphere samples were taxonomically more diverse than in the endosphere samples (*p* < 0.001, Welch Two Sample *t*-test, Figure 1B). For individual *H’* values per sample as well as Simpson’s indices (λ) and Evenness (*J´*) see Appendix A.

### 2.2. Compartment Rather Than Ontogenetic Stage Determines the Composition of Bean Prokaryotic Microbiota

A higher diversity of bacterial communities in the rhizosphere versus the endosphere has been reported, e.g., for rice, tomato, soybean, alfalfa, wheat, canola, poplar, pea, and lentil [4,5,23,65,71,72,73,74], although a study using sunflower demonstrated that this pattern, although common, is not ubiquitous [20]. As an independent way to visualize the community differentiation we used an unconstrained principal coordinate analysis (PCoA) of Bray-Curtis distances to quantify β-diversity and observed a clear separation of rhizosphere versus endosphere samples along the Axis 1 (explaining 92.5% of the overall variation), while the ontogenetic stage had a minor effect (Figure 2).

### 2.3. Dominant Taxa in Rhizosphere versus Endosphere

The bacterial community in the bean rhizosphere samples was dominated by six phyla: OTUs assigned to Proteobacteria (41%), Bacteroidetes (14%), Actinobacteria (13%), Gemmatimonadetes (6%), Chloroflexi (4%), and Acidobacteria (3.5%), altogether represented ~80% of the total number of classified sequences. In addition, 11.5% of the OTUs were assigned to the archaeal phylum Thaumarchaeota (Crenarcheaota). In the endosphere samples, over 99% of the sequences were annotated as Proteobacteria (Figure 3A). At the genus level, we could only classify ~41% and ~38% of the OTUs in the rhizosphere and endosphere samples, respectively. Dominant genera in the rhizosphere were *Candidatus Nitrososphaera* (11.5%), *Flavisolibacter* (3%), *Steroidobacter* (3%), *Kaistobacter* (2%), *Agrobacterium* (1%), and *Rubrobacter* (1%), while in the endosphere, most of the annotated sequences belonged to *Agrobacterium* resp. *Rhizobium* (each genus representing 30%), followed by *Ochrobactrum* (less than 1%) (Figure 3B). 

The beforementioned findings generally agree with those from other systems, reporting Proteobacteria, Acidobacteria, Bacteroidetes, and Actinobacteria as dominant phyla in the rhizosphere and further enrichment of Proteobacteria in the endosphere of diverse crops, including bean [4,9,10,16,23,24,25,37,59,61,72,75], although Firmicutes dominated among endophytic bacteria cultured from seeds or roots [52,53]. However, the high abundances of the phyla Chloroflexi and Gemmatimonadetes and genera such as *Flavisolibacter* or *Steroidobacter* and the archaea, *Candidatus Nitrososphaera*, appear to be less common features of common bean rhizosphere microbiota: OTUs assigned to the Chloroflexi or Gemmatimonadetes reached abundances of <0.5% and 1–1.5%, respectively, in the rhizosphere of wild and cultivated common beans grown in agricultural soil from Columbia [59], and similarly low abundances were reported for the bean plants grown in several agricultural regions in the USA [37]. Interestingly though, a study comparing two bean cultivars with different levels of resistance to *Fusarium oxysporum* identified Chloroflexi as a taxonomic group exclusively found in the rhizosphere of the resistant cultivar and reported higher abundances (ca 4%) in Amazon dark soil than in agricultural soil [60]. The only study reporting Gemmatimonadetes among the dominating phyla in the rhizosphere of common bean seems to be the work by Barraza et al. [61].

Gemmatimonadetes were among the four dominant phyla in the rhizosphere of lentil, pea, soybean, wheat, maize, canola, oilseed rape, artichoke, and diverse grassland species from temperate zones in Europe or the Chinese tundra [9,26,72,76,77,78,79,80]. Several of these studies demonstrated the enrichment of Gemmatimonadetes in the rhizosphere as compared to bulk soil [78] and linked this phylum to increased biomass production or increased resistance to salt stress [76,77,80], mainly due to abundant nitrogen-fixing *Gemmatimonas* spp. [9,71,78]. Moreover, among the same studies, high abundances of Chloroflexi were observed in the rhizosphere of soybean, oilseed rape, and wild grassland species [9,72,78], and—together with *Flavisolibacter*—even in the microbial community in the nodules of wild soybean [76]. OTUs assigned to the ammonia-oxidizing archaea *Candidatus Nitrososphaera* [81,82] were reported from the rhizosphere of oilseed rape and maize, and *Rhizobium* from the endosphere of oilseed rape, wheat, and canola roots [78,79]. All the latter studies share with our present work that plants were cultivated in the field, while the study by Barraza et al.—although using potted plants in a greenhouse—was performed in the same location as our work: that is, at least the airborne inoculum can be expected to be the same in both studies. Therefore, we conclude that N-fixing or ammonia oxidizing Gemmatimonadetes and archaea could represent candidates for functionally important components of the ‘original’ microbiome of bean that merit further investigation.

### 2.4. Ontogenetic Shift from Stenotrophomonas to N-Fixing Taxa in the Rhizosphere

Although we could not confirm a significant effect of the ontogenetic stage on the overall variation in the bacterial root microbiota in our ten samples (PERMANOVA *p* = 0.88, perm = 999), we detected 156 individual OTUs with significantly different abundances in pairwise comparisons between phenological stages: 154 OTUs (153 rhizosphere, 1 endosphere) differed between the vegetative and flowering stage, while only two OTUs differed between the flowering and pod filling stage (Figure 4).

Using a BLAST alignment against the NCBI nucleotide non-redundant database (22 July 2021) we could annotate 37 of the differential OTUs in the rhizosphere samples to genus level, dominating genera being *Rhizobium* (14 OTUs), *Aeromonas* (8 OTUs), *Stenotrophomonas* (4 OTUs) and *Paracoccus* (3 OTUs). The remaining 76% OTUs were classified as ‘uncultured bacterium’ (Appendix A). In spite of the low number of OTUs annotated to genus level, the reads assigned to these OTUs accounted for almost 97% of reads assigned to all 153 differential OTUs in the vegetative stage and for 85% in the flowering stage (corresponding to 1.5% and over 6% of the total read numbers, respectively). Interestingly, the only differential OTU in the endosphere was annotated as *Chryseobacterium* TS35. *Chryseobacterium* spp. are frequently reported from compost or other disease-suppressive soils, and several isolates exerted direct antibiosis via the production of chitinases or cellulases [12,83,84]. The very low abundance of this OTU evidently calls for cautious interpretations, but read numbers below 200 (corresponding to up to 0.5% relative abundance) also characterized OTUs identified as *Chryseobacterium* in the rhizosphere of wild genotypes and landraces—but not modern cultivars—of common bean [59]. Even more importantly, *Chryseobacterium* spp. can act as ‘hub’ species that facilitates the colonization of roots by other beneficial microorganisms [85], including the nod-factor-mediated colonization by *Rhizobia* [40].

Moreover, in line with a potential facilitating effect of *Chryseobacterium*, the abundance of OTUs annotated as *Rhizobium* exhibited a 40-fold increase (from 3000 to over 110,000 reads) from the vegetative to the flowering stage (equivalent to an increase from 11% to 83% of the differentially abundant reads) (Figure 4A, Table 2). A similar trend—although at much lower absolute values—was observed for *Aeromonas* (increasing from 11 to almost 800 reads) and *Paracoccus* (increasing from 14 to over 600 reads). By contrast, four OTUs classified as *Stenotrophomonas* showed the opposite trend and decreased from 24,000 to 1600 reads (Table 2). 

**Table 2 plants-11-01631-t002:** Rhizosphere OTUs with differential abundances between the vegetative and the flowering stage ^1^.

ID	Family	Genus	Read Numbers	
			Vegetative Stage	Flowering Stage
1083508	Xanthomonadaceae	*Stenotrophomonas*	18640	1298
537062	Xanthomonadaceae	*Stenotrophomonas*	3507	238
227343	Xanthomonadaceae	*Stenotrophomonas*	1926	110
4045633	Rhizobiaceae	*Rhizobium*	555	27149
843074	Rhizobiaceae	*Rhizobium*	465	12328
1104546	Rhizobiaceae	*Rhizobium*	401	12232
155854	Rhizobiaceae	*Rhizobium*	400	14421
714181	Rhizobiaceae	*Rhizobium*	365	18057
225582	Rhizobiaceae	*Rhizobium*	267	4970
1107243	Rhizobiaceae	*Rhizobium*	263	6907
80113	Rhizobiaceae	*Rhizobium*	179	8120
591708	Xanthomonadaceae	*Stenotrophomonas*	92	3
848768	Rhizobiaceae	*Rhizobium*	87	3650
220539	Rhizobiaceae	*Rhizobium*	79	1976
634321	Flavobacteriaceae	*Flavobacterium*	67	2
200464	Rhizobiaceae	*Rhizobium*	46	2024
1104627	Rhizobiaceae	*Rhizobium*	37	1685
370368	Rhodobacteraceae	*Paracoccus*	10	435
529216	Rhodobacteraceae	*Paracoccus*	4	103
833408	Aeromonadaceae	*Aeromonas*	3	203
38159	Rhizobiaceae	*Rhizobium*	3	78
834097	Aeromonadaceae	*Aeromonas*	2	129
1085832	Streptococcaceae	*Streptococcus*	2	94
813705	Aeromonadaceae	*Aeromonas*	2	88
831599	Aeromonadaceae	*Aeromonas*	1	151
837574	Aeromonadaceae	*Aeromonas*	1	60
423025	Aeromonadaceae	*Aeromonas*	1	54
564995	Rhizobiaceae	*Rhizobium*	1	51
1141678	Aeromonadaceae	*Aeromonas*	1	48
578911	Bacillaceae	*Exiguobacterium*	0	108
641892	Blastocatellaceae	*Aridibacter*	0	102
171996	Rhodobacteraceae	*Paracoccus*	0	89
830659	Streptococcaceae	*Lactococcus*	0	62
165293	Cyclobacteriaceae	*Algoriphagus*	0	58
388951	Aeromonadaceae	*Aeromonas*	0	55
1144093	Hyphomicrobiales	*Liberibacter*	0	41
1143479	Cyclobacteriaceae	*Algoriphagus*	0	38

^1^ This table presents absolute read numbers (sum of both replicates) of OTUs assigned to genus level, colour code ranges from dark blue (0 reads) over mauve (400) to red (highest read number). See Appendix A for a complete list of differential OTUs.

Ontogenetic effects on rhizosphere communities during the early stages of vegetative growth have been described, e.g., for soybean, maize, and rice [72,86,87,88]. In our study, changes in the abundances of few, but functionally important taxa resulted in a dominance shift from *Stenotrophomonas* in the vegetative stage to taxa like *Rhizobium, Aeromonas,* or *Paracoccus* spp. in the flowering stage. As mentioned for *Chryseobacterium* spp., *Aeromonas* spp. are frequently present in disease suppressing soils or compost [89], some shown to control soil-borne plant pathogens via chitinase secretion [90], and also *Paracoccus* comprises diverse beneficial rhizosphere bacteria [88], including seed endophytes of common bean [53]. *Stenotrophomonas* strains have been reported as leaf endophytes of common bean [55] or root endophytes of chickpea (*Cicer arietinum*) [91] and shown to exert antifungal activity against *Fusarium oxysporum,* and *S. maltophilia* is a commonly used biocontrol agent [92]. Generalizations concerning the putative effects are less straightforward for this genus than for, e.g., *Rhizobium*, because *Stenotrophomonas* isolates from soil or rhizosphere samples can comprise plant- and human pathogenic strains as well as plant-growth-promoting species [93,94,95]. Nevertheless, the taxonomic affiliations of the dominant OTUs indicate that the ontogenetic shifts in the common bean rhizosphere are likely to generate favourable effects for the plant, similar to a recent report on increasing abundances of growth-promoting bacteria over the ontogeny of the legume *Vigna subterranea* [96].

### 2.5. Factors Explaining the Differentiation between Endosphere and Rhizosphere Bacterial Communities Assembled from the Native Inoculum

#### 2.5.1. The Bean Root Endosphere Is Colonized by a Subset of the Rhizosphere Community

The observation that the strongest changes in root microbiota are usually associated with phases of particularly rapid growth and high N demand [37,86] is frequently interpreted as evidence that plants can select or ‘filter’ microbial communities according to their own, specific needs [4,86]. However, as in most published work, compartment rather than ontogeny explained a major part of the variation in our samples, and the factors and players that control the differentiation between rhizosphere and endosphere communities remain poorly understood. In principle—seed-born endophytes can serve as an inoculum of the rhizosphere [97], and PGPR from the genera Bacillus, Paenibacillus, Paracoccus, and—evidently—diverse rhizobia have been identified among the seed endophytes of common bean [53]. Our dataset characterizes the endosphere community as a subset of the rhizosphere community: we detected only six genera and no phylum among the taxonomic affiliations of the annotated OTUs that were exclusive to the endosphere (Figure 5, Appendix A). We consider this pattern to be more congruent with a model in which endosphere communities are mainly recruited from the rhizosphere.

#### 2.5.2. Asking “What Can They Do?” and “What Must They Do?” to Identify the Controlling Partner

The factors that act as the most important ‘filters’ in the structuring of rhizosphere versus endosphere microbiota remain a matter of discussion [98,99], but increasing consensus supports the relevance of bacterial function—rather than taxonomy [4,8,23]. As formulated by Liu & col (2017) [5], asking ‘what can they do in the endosphere?’ might be the most fruitful way to understand the composition of endosphere microbiota. However, this question intrinsically assumes that the plant can control the filtering process, and—as mentioned in the introduction—plants have less control in the rhizosphere. Since studies with a plant versus a microbiome focus do not necessarily yield the same results [100], we aimed to complement the approach suggested by Liu & col by asking ‘what must they do?’: that is, which functions are required by the bacteria to colonize the rhizosphere versus the endosphere. We argue that an overrepresentation of functions that mainly benefit the microorganism *versus* the plant in each compartment could reveal hints towards the partner that controls or modulates the ‘filtering process’. Therefore, we complied published information on functions/traits that characterise bacteria able to colonize the rhizosphere versus the endosphere, predicted functional gene profiles from our 16S rRNA gene sequence datasets, tested for the overrepresentation of selected key genes in each compartment, and finally, discuss these patterns from the perspective of each partner, asking ‘what they must do?’ for functions that benefit the microorganisms and asking ‘what can they do?’ for functions mainly representing as service for the plant.

#### 2.5.3. Things Bacteria MUST Do for Themselves and Things They CAN Do for the Plant

Many excellent reviews have analysed the most essential functions that bacteria require to reach and colonize the rhizosphere and eventually proceed to the inner parts and colonize the endosphere [4,5,6,101,102,103,104,105]. In short, ‘things bacteria MUST do’ in the rhizosphere include chemotaxis and motility, quorum sensing, biofilm formation, and adherence, to deal with defensive phytochemicals and diverse functions that enhance the degree of competitiveness, ranging from reaching high growth rates via the capacity to utilize diverse carbon sources or to successfully deal with nutrient depletion to direct antibiosis. Bacteria able to colonize the endosphere require functions that (i) allow for the adhesion to and proliferation of root surface structures, (ii) facilitate the eventual penetration of plant cell walls, and ultimately (iii) enable them to survive in spite of the plant immune systems; examples comprise adhesins and lipopolysaccharide formation, flagella and pili, twitching motility, quorum sensing, the production of cell-wall degrading enzymes, the detoxification of ROS and factors for the suppression and/or evasion of other plant immune responses. Correspondingly, a metagenomic study of the bacterial endosymbionts of rice roots revealed numerous copies of genes encoding for plant-polymer-degrading enzymes, such as cellulases and xylanases, or being involved in the detoxification of reactive oxygen species (ROS), all components of the flagellar apparatus and—with exception of the type III secretion system—all components of all known protein secretion systems [106].

As shortly mentioned in the introduction, some, but not all, of these functions intrinsically generate plant-beneficial effects, examples comprise traits related to antibiosis, microbial competition, and nutrient mobilization [4,6,7,101,107,108]. However, it seems reasonable to assume that bacterial mobility, transporter systems, and the capacity of plant cell wall degradation are things bacteria ‘must’ do in the respective compartment and hence, in the first line benefit the bacteria. By contrast, N fixation passes by far the quantities required for the bacterial metabolism and thus, is usually considered a ‘service’ for the plant.

#### 2.5.4. Things Bacteria MUST Decide Who Dominates the Bean Rhizosphere

We used PICRUST [66] and the PICRUSt2 QIIME2 plugin to predict metagenome functions from our 16S rRNA gene sequences [109,110] and to identify Kyoto Encyclopedia of Genes and Genomes (KEGG) orthologs (KOs) (https://www.genome.jp/brite/ko00001, accessed on 1 April 2000) that appeared particularly enriched in a specific compartment. We identified 64 KOs, most of which belonging to membrane transport, amino acid metabolism, carbohydrate metabolism, and replication and repair, as the most abundant categories in both compartments. Perhaps not surprisingly, we detected the strongest differences (difference in mean populations = ~8%, q-value = 3.33 × 10^−5^) between the compartments for the abundances of predicted genes in KOs belonging to Environmental Information Processing, Chemotaxis, and Motility, and Membrane Transport, particularly ATP-binding cassette transporters (ABC-transporters), general transporters, phosphotransferase system (PTS), and bacterial secretion systems such as type IV secretion system and type VI secretion systems (T4SS and T6SS, respectively). Interestingly though, within each KO, we detected predicted functions that were enriched in each of the compartments. For example, among functions belonging to chemotaxis and motility (or signal transduction, respectively), aspartate chemotaxis and galactose chemotaxis were enriched in the rhizosphere, while serine chemotaxis, ribose chemotaxis, and flagellum apparatus were enriched in the endosphere. Among the secretion systems, predicted genes belonging to Type I pili, Type II, III and Type VI secretion systems as well as twitching motility were enriched in the rhizosphere, while a Type IV SS component was enriched in the endosphere (see Table 3 for selected key genes). Similarly, among the predicted ABC transporters, those for capsular polysaccharide, spermidine/putrescine, L-cystine, histidine, lysine/arginine/ornithine, and rhamnose protein were enriched in the rhizosphere, while ABC-transported for dipeptide, branched-chain amino acid, methionine, and L-arabinose were enriched in the endosphere.

The observed enrichments per compartment followed the expected patterns in most, but not all cases. Evidently, the reliability of functions predicted from 16S-amplicons depends on the degree of phylogenetic conservation of each trait, and thus, any interpretation remains speculative [111]. However, functions that require interaction among multiple proteins tend to be phylogenetically conserved [111], and we successfully validated our predictions using degenerate primers for a PCR-based amplification from the original DNA for five key genes, which were selected to represent different KOs: general secretion pathway protein D (gspD) as part of the type II secretion system, motility protein A (motA) as representative of chemotaxis, butanediol dehydrogenase (butB) as part of the biosynthetic pathway for a particularly important volatile component of host-microbe signaling, N-acetylglucosamine (GlcNAc) transferase as representative of a nodulation (nod) gene, and the nitrogenase, nitrogen-fixing H (nifH), as a key step in bacterial nitrogen fixation (see Appendix A). Therefore, we assume that particularly functions related to nitrogen cycling or fixation, bacterial secretion systems, and motility are likely to be reliably predicted. In the following, we discuss examples of predicted functions for which we detected a significant enrichment in a compartment, compare the observed to the expected patterns and interpret our observations with an emphasis on the partner these functions should benefit most (Table 3).

**Table 3 plants-11-01631-t003:** Predicted genes with beneficial effects for the bacterium, the plant, or both partners: observed and expected enrichment in the rhizosphere versus endosphere ^1^.

Functional Group/KO	Gene	KEGG ID	Sign.	Enrichment		Comment	Ref
				O	E		
**Beneficial for bacterium**							
**Degradation of plant polymers**							
Endoglucanase	-	K01179	***			Required to penetrate root and cell surfaces and—eventually—for the liberation of nutrients from abundant plant structural molecules	[5,92,106]
Endo-1,3(4)-β-glucanase	-	K01180	*				
Endo-1,4-β-xylanase	xynA	K01181	*				
Oligo-1,6-glucosidase	malL	K01182	***				
Polygalacturonase	-	K01184	**				
Xylan 1,4-β-xylosidase	xynB	K01198	***				
Licheninase	bglS	K01216	***				
**Stress-related enzymes**							
Glutathione peroxidase	btuE	K00432	***			Detoxification of ROS	[106]
Glutathione S-transferase	gst	K00799	***				
Catalase	katE	K03781	**				
**Secretion systems**							
Type II	gspD	K02453	***			Injection of effectors into eukaryotic host cells to suppress host immunity	[106,112,113]
Type IV	virB2	K03197	***				
Type VI	hcp	K05601	***				
Type I pilus assembly	fimA	K07345	**				
**Chemotaxis and motility**							
Chemotaxis protein	motA	K02556	**			General chemotaxis	
Serine	tsr	K05874	**			Carbon source chemotaxis	[92]
Aspartate/maltose	tar	K05875	***				
Ribose	rbsB	K10439	***				
Galactose	mglB	K10540	***				
Flagellar apparatus	fliI	K02412	(*)			Overrepresented in rice endosphere	[5,106]
Twitching motility	chpA	K06596	***			Usually considered relevant to colonize the endosphere	[6,114]
	pilJ	K02660	***				
**Signal trans. 2-comp systems**							
Carbon source utilization	creC	K07641	**			Carbon source utilization and toxin resistance required for rhizosphere colonization	
	creB	K07663	***				
Multidrug resistance	baeS	K07642	***				[4,6]
	baeR	K07664	***				
Antibiotic resistance	evgS	K07679	***				
	evgA	K07690	***				
Amino sugar metabolism	glrK	K07711	***				
**Cell fate control**	pleD	K02488	***				[115]
	pleC	K07716	***				
**Beneficial for both**							
**Plant growth promotion**							
Acetoin reductase	budC	K03366	***			Synthesis of 2,3-butanediol, VOC, growth and resistance induction, bacterial survival	[116,117,118]
Acetolactate decarboxylase	alsD	K01575	***				
Butanediol dehydrogenase	butB	K00004	***				
**Plant hormones**							
S-adenosylmethionine syt.	metK	K00789	***			Ethylene for suppression of plant resistance, indole acetic acid (IAA) enhances bacterial rhizosphere competence	[5,92]
IAA biosyn. IAM pathway	amiE	K01426	***				[119,120]
IAA biosyn. IPyA pathway	ipdC	K04103	***				
**Degradation of microbial polymers**							
Chitinase	-	K01183	***			Antibiosis, microbe-microbe competition, or ‘biocontrol’ of microbial pathogens	[4,6]
Chitinase/lysozyme	chiA	K13381	***				
**Service for plant**							
**nod genes and nitrogen fixation**							
LysR family TF	nodD	K14657	***			Nod genes required for plant colonization and nodule formation, nifA controls the nif operon, nitrogenase nifH, for nitrogen fixation, 2 component system proteins ntrY and ntrX control nitrogen fixation and metabolism	[92]
Nodulation protein	nodA	K14658	***				
Chitooligos-deacetyl	nodB	K14659	***				
nodulation protein	nodE	K14660	***				
	nodF	K14661	***				
GlcNAc transferase	nodC	K14666	***				
Nif regulatory protein	nifA	K02584	***				
Nitrogenase	nifH	K02588	ns				
Histidine kinase	ntrY	K13598	***				
N regul. response factor	ntrX	K13599	***				
**Nutrient solubilizing**							
3-phytase	-	K01083	*				

^1^ Predicted functional genes for which we detected a significant enrichment are grouped according to their Kyoto Encyclopedia of Genes and Genomes (KEGG) Orthology, (KO) terms. Asterisks indicate significance levels according to Welch’s t-test (* *p* < 0.05, ** *p* < 0.01, *** *p* < 0.001, (*) *p* = 0.086, ns *p* > 0.05, *p*-values adjusted for multiple comparisons with the FDR correction using the Benjamin-Hochberg method). Colours illustrate the observed (O) and expected (E) enrichment in the rhizosphere (green-cyan) or endosphere (orange) respectively. See Appendix A for details.

We detected several functions that called our attention because traits usually considered as a requirement of endosphere bacteria appeared to be overrepresented in the rhizosphere and vice-versa (Table 3). For example, the predicted rhizosphere communities were enriched in enzymes that degrade plant-derived macromolecules, including endo-glucanases, endo-1,4-β-xylanase, poly-galacturonase, and licheninase. These functions are required to penetrate plant cell walls and thus, they represent key traits of successful endosphere bacteria that do not provide any direct benefit to the plant. In the rhizosphere, however, enrichment of these functions could result in an increased availability of nutrients that favours both the plant and the rhizobacteria. Similarly, bacterial secretion systems allow for the injection of effectors into eukaryotic cells generally considered essential components of the genomes of pathogens, although the priming of host immunity [106,112,113] can generate a ‘secondary’ benefit for the plant. Although these functions should lead to an enrichment of secretion systems in endophytic communities, we observed the opposite pattern for several predicted components. If confirmed (See Appendix A), this pattern could indicate a contribution of secretion systems to the successful colonization of the rhizosphere. By contrast, two predicted genes required for the synthesis of 2,3-butanediol—a well-known mediator of airborne signalling—were overrepresented in the endosphere, a pattern that we could validate for butB (see Appendix A). The volatile 2,3-butanediol represents a common PGPR-derived signal that triggers growth promotion and resistance induction effects without physical contact between the bacteria and the root. Moreover, recent work shows that 2,3-butanediol can enhance the rhizosphere competence of the emitting strains [116,117,118]. An enrichment of its biosynthetic pathway in endosphere communities could indicate an as-yet seldomly considered ‘microbial small talk’ sensu Ryu and colleagues [121] that is taking place inside the plant. Similarly, two predicted enzymes for the formation of cyclic di-3′,5′-guanylate (c-di-GMP), di-guanylate cyclase (pleD), and its cognate receptor-like kinase, pleC, were significantly overrepresented in the endosphere. The second messenger c-di-GMP is a ubiquitous bacterial second messenger that controls a plethora of cellular processes determining bacterial lifestyle switches that are related to environmental adaptations, including biofilm formation, dispersal, and motility [115,122,123]. The diversity of processes controlled by c-di-GMP and the continuum from mutualistic to pathogenic outcomes makes any generalization impossible, but at the very least, the overrepresentation of pleC and pleD in the endosphere underlines the relevance of these processes for the colonization of this compartment. Finally, nitrogen fixation is considered the most important microbial service for plants that are provided by nodule-forming taxa, at least in legumes. Thus, genes involved in the nodulation process or the subsequent assimilation of nitrogen should be enriched in the endosphere. Indeed, we observed an enrichment of predicted nod genes in the endosphere which could be validated for nodC. By contrast, nitrogenase (nifH) occurred at similar abundances in the predicted functions in both compartments. In summary, we detected several cases of overrepresented genes among the functions predicted from our amplicon data which might indicate an unexpected representation of plant-beneficial bacteria in the rhizosphere and—to a lower extent—the endosphere of bean plants that grow within their native habitat range.

## 3. Materials and Methods

### 3.1. Plant Material, Growing Conditions, and Sampling

Seeds of the *Phaseolus vulgaris* cultivar Flor de Junio Marcela were kindly donated by Dr. Jorge Acosta from the national germplasm collection of Instituto Nacional de Investigaciones Forestales Agrícolas y Pecuarias (INIFAP), Celaya, GTO, México. The field plot was located within an experimental field at CINVESTAV—Irapuato that has been used to cultivate different wild and domesticated genotypes of common bean without the use of any type of pesticides during the past eight years. The geographic location (1800 m above sea level; 20°43′13″ N; 101°19′43″ W) in the Bajio region in Central Mexico (state of Guanajuato) falls within the area considered the Mesoamerican domestication centre of common bean and the diversity centre of wild *P. vulgaris* genotypes [43,47,64]. Moreover, the cultivar Flor de Junio Marcela has been mass-selected for disease resistance, plant vigour, pod load etc., in experimental fields within the same region [124,125], and many smallholder and rural farmers in the Bajio regularly cultivate common bean, with Flor de Junio beans being among the most preferred cultivars [64].

A total of 42 seeds were surface-sterilized and subsequently sown directly into the soil, timing of sowing (in May, the beginning of the rainy season) and culturing methods followed the conditions for growing bean in the region, with the exception that no pesticides or fertilizers were used. At each of the three selected phenological stages (‘vegetative stage’: appearance of the 3rd trifoliated leaf, ‘flowering’: appearance of the first open flowers, ‘early pod filling’: first pods of at least 1 cm length, corresponding to stages 2-4 as described in [37]), we randomly selected three individual plants to collect rhizosphere and endosphere samples following [65,126]. This sampling effort was performed in duplicate for the first two stages, but not the last stage, yielding a total of ten samples (see Appendix A). Upper organs of the plant were cut off and—using a flame-sterilized scalpel—the soil attached to the root was removed mechanically and collected in a 1.5 mL Eppendorf tube, yielding approximately 1 g of rhizosphere sample per plant. Subsequently, roots were placed in a sterile 50 mL Falcon tube with 25 mL of sterile phosphate buffer (0.1 M, pH 8), and vortexed for 15 s at maximum speed to remove most of the remaining soil attached to the root. These steps were repeated until the buffer remained clean (without sediment on the bottom) and subsequently, the roots were rinsed 3 times with sterile distilled water. Subsequently, the roots were transferred into a new 50 mL Falcon tube with 25 mL of 70% (*v*/*v*) ethanol for 1 min, followed by 25 mL of 5% (*v*/*v*) hypochlorite sodium (NaClO) where the roots rested for 5 min. These washes were repeated two more times. One last series of three washes with sterile distilled water was carried out, and then roots were frozen and stored at −80 °C. As a control for successful surface sterilization, the water used for the last series of washes was plated on Petri dishes with LB solid medium and incubated at 28 °C for three days, and only bean roots from which no visible bacterial colonies could be cultured were conserved for further analyses.

### 3.2. DNA Extraction from Soil (Rhizosphere) and Root (Endosphere) Samples

Total DNA was extracted from 0.2–0.3 g rhizosphere soil per plant using the MoBio PowerSoil^TM^ DNA Isolation Kit (MoBio Laboratories, Inc., Carlsbad, CA, USA) following the manufacturer’s instructions, but using 30 µL of distilled sterile water for the last elution. To extract DNA from the root endosphere samples, the ‘hot CTAB DNA isolation method’ was used as previously reported [127] but with some modifications. Nodules were removed with a flame-sterilized scalpel from the sterilized roots and subsequently, 1 g of root tissue was frozen with liquid nitrogen and ground with a mortar and pestle, then transferred to a 1ml Eppendorf tube and suspended in 300 µL of extraction buffer [2% (*w*/*v*) cetyltrime-thylammonium bromide (CTAB), 100 mM Tris-HCl (pH 8.0), 1.4 M NaCl, and 20 mM EDTA], mixed by inverting the tube several times and incubated for 10 min at 65 °C and for further 10–20 min at room temperature. After adding 300 µL of chloroform and mixing vigorously, the suspension was centrifuged for 5 min at 10,000× *g*, the aqueous phase was transferred to a new sterile 1 mL Eppendorf tube, mixed with 300 µL of isopropanol and vigorously shaken on a vortex. Total DNA was precipitated via centrifugation (5 min at 10,000× *g*), the supernatant was discarded, and the pellet was washed with 500 µL of 70% (*v*/*v*) ethanol. Finally, the ethanol was discarded, and the pellet was air-dried and suspended in 100 µL of distilled sterile water. The purity and quality of the total DNA extracted from both compartments was checked individually for each sample by electrophoresis on a 1.1% (*w*/*v*) agarose gel and using a Nanodrop 2000 (Thermo Scientific, Wilmington, DE, USA).

### 3.3. Generation of 16S rRNA Gene Amplicons

As mentioned in the Results Section 2.1, the bacterial 16S rRNA gene was amplified from rhizosphere and endosphere DNA using the primers 799f (AAC MGG ATT AGA TAC CCG G) and 1492r (TAC GGY TAC CTT GTT ACG ACT T), targeting the V5–V9 hypervariable region which should give a product of approximately 750 bp [68]. To determine the successful size-dependent separation, PCR using these primers and total DNA from endosphere samples were performed and the product was separated by electrophoresis agarose gel. The PCR yielded two bands of approximately 1300 bp and 750 bp, respectively. Both bands were excised, purified using the Illustra GFX DNA and Gel Band Purification Kit (GE Healthcare, Amersham Place, LC, UK), and the resulting amplicon sequences were cloned, following the preparation and transformation of chemically competent *E. coli* DH5α cells as described by Sambrook and Russel [128]. Ligation of the DNA fragments was performed using the pJET1.2/blunt-cloning vector from the CloneJET PCR Cloning Kit (ThermoScientific, Wilmington, DE, USA) following the manufacturer’s instructions. Once positive colonies were selected, plasmidic DNA was extracted using published protocols [128] and used as a template for PCR. The 25 µL PCR reaction mixture contained approximately 100 ng of plasmidic DNA, 10 × PCR reaction buffer, MgCl_2_ 2 mM, 10 pmol of each primer (pJET1.2 forward 5′-CGA CTC ACT ATA GGG AGA GCG GC-3′ and pJET1.2 reverse 5′-AAG AAC ATC GAT TTT CCA TGG CAG-3′), 100 µM of each dNTP, and 1.5U of Taq DNA Polymerase (Invitrogen). After initial denaturation at 95 °C for 3 min, each thermal cycle was performed as follows: denaturation at 94 °C for 30 sec, annealing at 60 °C for 30 s, and elongation at 72 °C for 1 min. At the end of 30 cycles, the final extension step was at 72 °C for 8 min. Twenty-eight positive clones were sequences by the Sanger method at the National Laboratory of Genomics for Biodiversity (LANGEBIO), at Centro de Investigación y de Estudios Avanzados (CINVESTAV)—Unidad de Genómica Avanzada, Irapuato, GTO, México (https://langebio.cinvestav.mx/).

Since BLAST analyses unambiguously confirmed that the sequences of all 750 bp products matched to bacterial 16S rRNA genes and those of the 1300 bp products to *Phaseolus* mitochondrial genes, we used the beforementioned primers to generate amplicons from all ten samples. The 25 µL PCR reaction mixture contained approximately 1 ng of total DNA for rhizosphere samples and 10 ng for root endosphere samples, 10× PCR reaction buffer, MgCl_2_ 2 mM, 10 pmol of each primer, 100 µM each dNTP, and 1.5U of Taq DNA Polymerase (Invitrogen). After initial denaturation at 94 °C for 5 min, each thermal cycle was performed as follows: denaturation at 94 °C for 1 min, annealing at 52 °C for 45 sec, and elongation at 72 °C for 1 min. At the end of 30 cycles, the final extension step was at 72 °C for 8 min. For rhizosphere samples, six parallel reactions per sample were performed, PCR products were purified and concentrated, using the Illustra GFX DNA and Gel Band Purification Kit (GE Healthcare, Amersham Place, LC, UK) and pooled. In the case of the endosphere samples, PCR products from six parallel reactions per sample were pooled, separated by electrophoresis on a 1.1% agarose gel, and the 750 bp band was excised and purified using the Illustra GFX DNA and Gel Band Purification Kit. High-throughput sequencing of the resulting 16S rRNA gene amplicon pools was performed on the MiSeq 2 × 300 Illumina platform.

### 3.4. Bioinformatic Data Processing and OTU Picking

Bioinformatic processing and data analysis followed general protocols [129], including using the Quantitative Insights into Microbial Ecology (QIIME) pipeline [70] for quality filtering, processing of the amplicons, and most subsequent analyses. However, aiming to preserve reads with at least a certain length, we used Trimmomatic version 0.33 [130] as a ‘pre-filter’ (filtering parameters: ILLUMINACLIP:TruSeq3-PE.fa: 2:30:10 LEADING:3, TRAILING:3 SLIDINGWINDOW:10:20 MINLEN:150) before using QIIME for quality filtering and the removal of barcode and primer sequences. Pre-processing parameters were adjusted in order to keep a larger quantity of sequences without causing an impact on the taxonomic distribution of the samples, as suggested by Bokulich et al. [131]. OTU determination was performed using a closed-reference approach at a 97% identity level (conventionally assumed to represent bacterial species [132]) using the greengenes gg_13_8_97 database as a reference and QIIME default parameters, with the option of reverse strand matching, enabled. RTAX [133] was used for taxonomy assignment, an algorithm using the information of both reads to reach more accurate annotations in the case of short, non-overlapping paired sequences. Subsequently, an OTU abundance table was constructed and the phyloseq version 1.16.2 [134] R package was used for all the downstream analyses.

### 3.5. Rarefaction Curves and Diversity Analysis

Rarefaction curves were constructed using an adaptation of the MacQIIME (version 1.9.0) command multiple_rarefactions.py in the R phyloseq 1.16.2 version package [134] from (https://github.com/joey711/phyloseq/issues/143, accessed on 6 August 2015), using the following parameters: lowest rarefaction depth in the series of depths (m), highest rarefaction depth in the series of depths (x), highest rarefaction depth in the series of depths (s), step size to increment from the low to high depths and the number of replicates to perform at each depth (n); where m = 1000 reads/sample, x = 1,100,000 reads/sample, s = 1000 reads/sample and n = 3 replicates. The mean values resulting from the three replicates at each step were used for constructing the rarefaction curves of every sample. To visualize the diversity structure of all sample communities, Principal Component Analysis was performed. The Bray-Curtis dissimilarity (distance) metric was calculated with the phyloseq R package [134]. Venn Diagrams were constructed based on the absolute numbers of taxa shown in Appendix A and the heatmap shown in Table 2 was constructed in Excel using a three-level colour code from blue (RGB: 68, 114, 196) for the lowest value, pink (RGB: 240, 148, 187) for the value 400 and red (RGB: 255, 0, 0) for the highest value.

### 3.6. Statistical Analysis and Differential Abundances

To determine OTUs appearing at different abundances between ontogenetic stages, the OTU abundance table generated by QIIME was analysed using the Statistical Analysis of Metagenomic Profiles (STAMP) software package was used [135]. *P*-values were calculated using the Welch’s t-test, while confidence intervals were calculated using the Welch′s inverted test for confidence interval method. *p*-values < 0.05 were considered as significant. Differential abundance analyses were performed using the Bioconductor package edgeR v3.30.3 [136]. We considered the OTUs with a >2-fold change between phenological stage samples and *p*-value < 0.05 (FDR-corrected) to be the enriched community. All *p*-values were adjusted for multiple comparisons with the FDR correction using the Benjamin-Hochberg method.

### 3.7. Prediction of Functional Genes

We used PICRUST [66] and the PICRUSt2 QIIME2 plugin [109,110] to predict functional gene profiles from the 16S rRNA gene sequences and to analyse the predicted metagenomic functions in our samples for the Kyoto Encyclopedia of Genes and Genomes (KEGG) orthologs (KOs) (https://www.genome.jp/brite/ko00001, accessed on 1 April 2021) which appeared particularly enriched in each of the compartments.

To validate these predictions, we selected five genes from different functional groups for PCR-based amplification from the original total DNA used for amplicon generation. Published degenerate primers were used for niH [137] and nodC [138], while primers for motA, butB, and gspD were designed de novo using ARDEP [139], following the instructions of the authors on the webpage (ijerph-17-05958.pdf (rcees.ac.cn) accessed on 5 February 2022). See Appendix A for primer sequences and Appendix A for detailed PCR conditions used for each gene.

## Figures and Tables

**Figure 1 plants-11-01631-f001:**
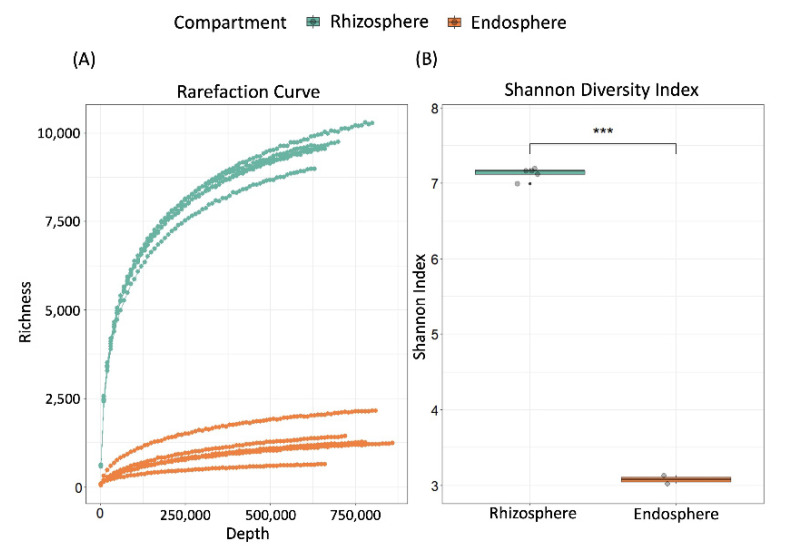
Alpha diversity of the bacterial microbiota in the endosphere and rhizosphere of common bean (*P. vulgaris*) grown in situ in the open field. (**A**) Rarefaction curves of the numbers of operational taxonomic units (OTUs) observed in the rhizosphere (green-cyan) and endosphere (orange) samples were calculated individually for each sample at an increment from the lowest depth (1000 reads) to the highest depth (approx. 1 million reads), each point represents the mean of three mathematical replicates. (**B**) Average Shannon diversity (*H’*) per compartment, asterisks *** indicate a statistically significant difference between compartments (*p* < 0.001, Welch Two Sample *t*-test, *n* = 5 biologically independent replicates).

**Figure 2 plants-11-01631-f002:**
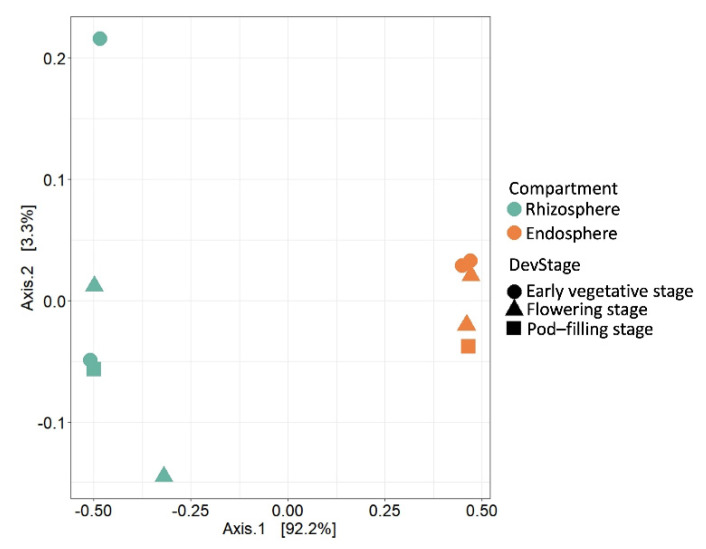
Community structure of the root microbiome of field-grown common bean (*P. vulgaris*) at three ontogenetic stages. Principal Coordinate Analysis (PCoA) of 16S rRNA diversity of rhizosphere (green-cyan) and endosphere (orange) samples taken at the three developmental stages (DevStage) indicated by symbol forms as early vegetative (circles), flowering (triangle) or pod-filling stage (squares). For each compartment, *n* = 2 biologically independent samples for the vegetative and flowering stage and *n* = 1 for the pod-filling stage.

**Figure 3 plants-11-01631-f003:**
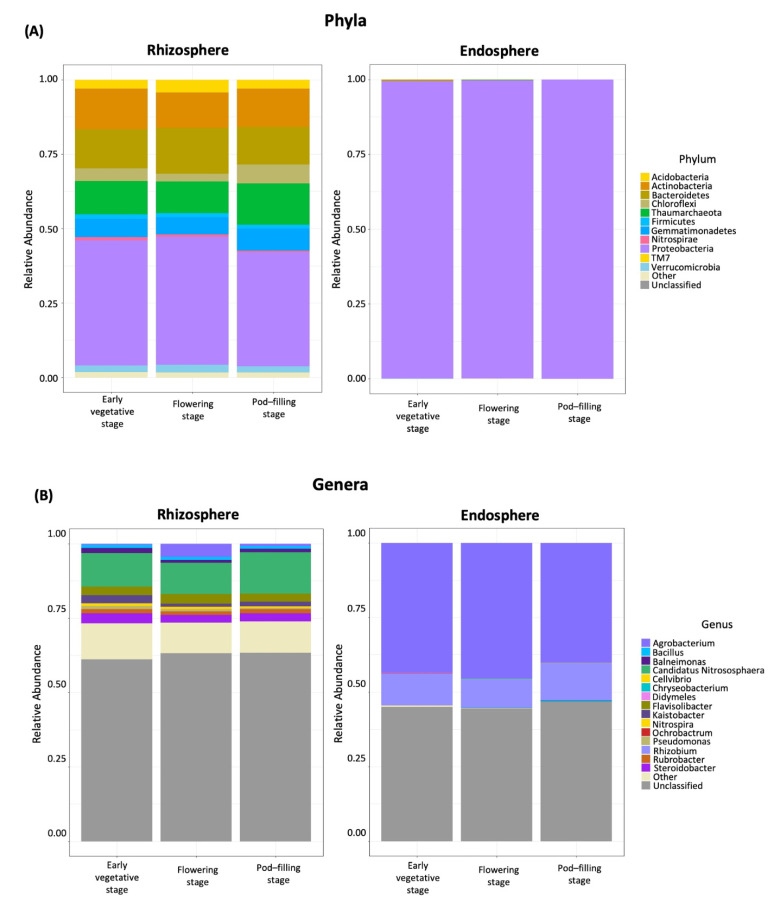
Dominating bacterial (and archaeal) taxa in the root microbiome of common bean (*P. vulgaris*) at three ontogenetic stages. The relative abundance of the ten most abundant taxa classified at the level of phylum (**A**) and genus (**B**) among all operational taxonomic units (OTUs) is presented for rhizosphere (**left**) and endosphere (**right**) samples separately for the vegetative, flowering and pod filling stage. Each bar plot represents data of *n* = 2 biologically independent samples for the vegetative and the flowering stage and *n* = 1 for the pod-filling stage and each colour indicates a different phylum or genus.

**Figure 4 plants-11-01631-f004:**
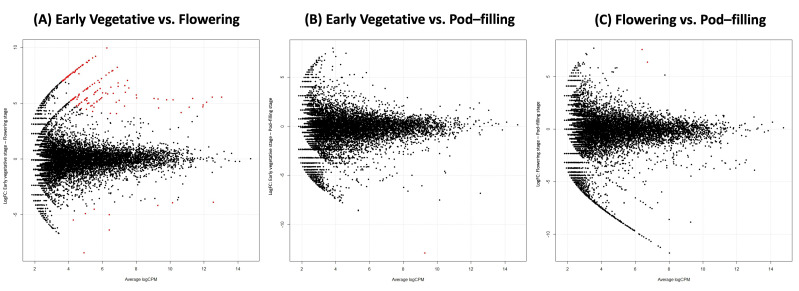
Differentially abundant operational taxonomic units (OTUs) in the rhizosphere of field-grown common bean (*P. vulgaris*) plants at three ontogenetic stages. The plots indicate the average log CPM (counts per million) and the log2 FC (fold change) of all OTUs comparing (**A**) the vegetative versus the flowering stage (**B**) vegetative versus pod filling stage and (**C**) flowering versus pod-filling stage, red dots indicate the significantly enriched OTUs (fold change > 2 and *p* < 0.05, FDR corrected).

**Figure 5 plants-11-01631-f005:**
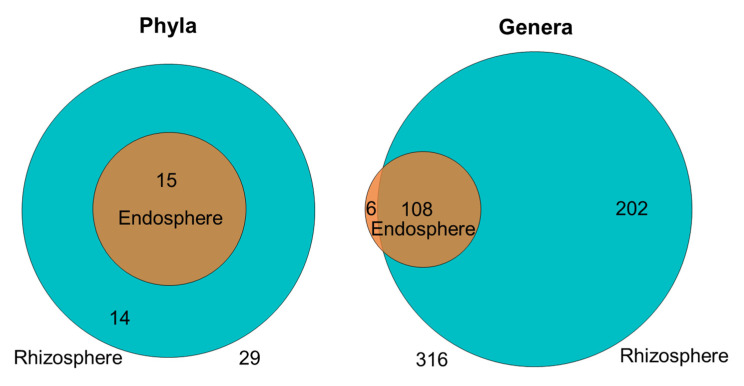
Distribution of shared and unique taxa by compartment. The diameters of the circles illustrate the number phyla (**left**) and genera (**right**) in each compartment (green-cyan, rhizosphere; orange, endosphere), numbers of unique taxa shown inside circles, numbers of shared taxa shown in overlapping areas, total numbers shown below the Venn Diagram.

## Data Availability

The datasets generated during the current study are available at NCBI (https://www.ncbi.nlm.nih.gov/bioproject/, accessed on 1 April 2021 and https://www.ncbi.nlm.nih.gov/nuccore/, accessed on 1 April 2021), as BioProject ID: PRJNA783305, BioSample accessions SAMN23426372, SAMN23426373, SAMN23426374, SAMN23426375, SAMN23426376, SAMN23426377, SAMN23426378, SAMN23426379, SAMN23426380, SAMN23426381, submission date 24 November 2021. For accession numbers of clones from the two bands of PCR products see Appendix A. R-scripts are available in GitHub (github.com) as frankmed/PAPER_Medina_2021_AllSetBeforeFlowering.

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
