# Peer review of "All Set before Flowering: A 16S Gene Amplicon-Based Analysis of the Root Microbiome Recruited by Common Bean (Phaseolus vulgaris) in Its Centre of Domestication"

_plants, 2022, doi:10.3390/plants11131631_

Round 1

Reviewer 1 Report

In the article entitled, “All set before flowering: A 16S gene amplicon-based analysis of the root microbiome recruited by common bean (Phaseolus vulgaris) in its centre of domestication” submitted by Martin Heil, authors have grown at the centre of domestication to characterise rhizospheric and endospheric bacterial communities at various stages of plants from vegetative to reproductive tissue. They found that Rhizosphere communities were four times more diverse than in the  endosphere and dominated by Actinobacteria, Bacteroidetes, Crenarchaeota and Proteobacteria and many other analyses. The study is nicely designed and executed, but it needs certain improvement as suggested-

  1. As already mentioned by authors that there are several similar studies have been done in the same plant (Table 1), then why there was a need an additional study? How your study is different from others?
  2. Results and discussion should be written separately as per the journal format.
  3. Besides 16S, is there any other method for the analysis of microbial community?
  4. Metagenome analysis could provide better results about the community than the 16S sequences, why it was not included?
  5. Is there any significant change in microbial community during various year of analysis? How and why?
  6. Table 3. Authors did only 16S sequencing, then how you could determine these genes and their functions? As per the analyses, these genes are reported from the related bacteria, but that doesn’t mean they are present and active at the site of study also?

These could be confirmed by metatranscriptome or metagenome analysis.

Or Atleast, author should confirm the presence of these gene by amplification from the DNA samples isolated from the site, otherwise it doesn’t mean anything.

  1. There are several typographical and grammatical error which needs to be corrected.
  2. Use uniform system of naming plants either using scientific name or common name when mentioning together e.g., in line no. 74.
  3. Few of the lines mentioned below should be checked and corrected-
  • Line no. 54-60: This paragraph is too long, please break into 2-3 sentences.
  • Line no. 74: “ex-situ” should be in italic.
  • Line no. 75-80: Sentence is long and unclear.
  • Line no. 105: If Phaseolus vulgaris is mentioned already it should be written as vulgaris.
  • Line no. 119-122: Sentence is unclear.
  • Line no. 150-155: Paragraph is long and confusing.
  • Line no. 172: “major” is spelled incorrectly
  • Line no. 168-160: Rephrasing is required

Author Response

We thank referee 1 for the general opinion that “The study is nicely designed and executed, but it needs certain improvement as suggested” and we hope that the referee will be satisfied with the way in which we made use of her/his suggestions.

General comments 1, 3 and 4. Among the general comments, the referee asked how our study differs from published work and why we choose 16S sequencing among the diverse methods available.

We added more information on the potential importance of locally distributed bacterial taxa (e.g., lines 90-98) and more clearly identify these taxa – which most likely had been missed by the earlier studies – as the major target of our work (lines 96-98 and 144-150). Furthermore, we now mention some of the most important methods available and shortly discuss their advantages and shortcomings, to justify our selection of 16S sequencing (lines 151-165). Thanks to a significant restructuring of the introduction section, we managed to add all this information at a very moderate increase in the length of the introduction (from 1500 words in the original version to ca 1650 words).

Comment 2. “Results and discussion should be written separately as per the journal format.”

Reply: The journal explicitly allows to combine these two sections and – after a carful consideration of the option mentioned by the referee – we still think that the combined form allows for a better ‘story-telling’.

Comment 5 “Is there any significant change in microbial community during various year of analysis? How and why?”

Reply: Since we obtained data during a single year, discussing any change over years would remain merely speculative.

In Comment 6, the referee expressed doubts concerning the prediction of functional genes from 16S amplicons, although these functions could be confirmed by metatranscriptome or metagenome analysis and added the suggestion to confirm the presence of these gene by amplification from the DNA samples isolated from the site.

Reply: We thank the referee for this criticism and in response, explain the main rationales for choosing this option to predict gene functions in the discussion section lines 151-165. In addition, we followed the suggestion and performed further experimental work to validate the presence and predicted patterns by PCR amplification of selected key genes from the original DNA. For the sake of time and since degenerate primers had to be designed de novo for most cases, we limited this new work to five candidate genes, which were selected for their importance in the particular context of the present study. The validation approach including the chosen key genes is mentioned just before citing Table 3 (lines 498-506), its results form part of the new Figure S1 and are mentioned for each key gene as part of the respective discussion. The  methods are summarized in  section 3.6., Primer sequences are presented in the new Table S6 and PCR conditions are reported in supplementary Text S1. . We hope that the referee sees and values this effort and thank again for a comment that has added a relevant new component to this work. 

Detailed comments

There are several typographical and grammatical error which needs to be corrected.

Use uniform system of naming plants either using scientific name or common name when mentioning together e.g., in line no. 74.

Few of the lines mentioned below should be checked and corrected-

Reply Au: Done

  • Line no. 54-60: This paragraph is too long, please break into 2-3 sentences.

Reply Au: Done, see new version lines 50-59.

  • Line no. 74: “ex-situ” should be in italic. Done (now line 73)
  • Line no. 75-80: Sentence is long and unclear. This sentences has been eliminated
  • Line no. 105: If Phaseolus vulgaris is mentioned already it should be written as vulgaris. We agree that the referee is technically correct. Since we consider is important to indicate the full Latin name in the place where this species is identified as the model of the present work, we avoided mentioning the full species name earlier in the introduction
  • Line no. 119-122: Sentence is unclear. We agree and hope that we could resolve this issue (new version lines 114/115)
  • Line no. 150-155: Paragraph is long and confusing. The sentence has now been broken down in two (lines 167-171)
  • Line no. 172: “major” is spelled incorrectly. Corrected
  • Line no. 168-160: Rephrasing is required. Done

Reviewer 2 Report

- In general terms the topic of the article is interesting, the methodology is explicitly presented and the results reported are interesting. 

- In my opinion, the introduction chapter should end with a paragraph indicating the purposefulness of the conducted research. Authors should clearly define the purpose of the work and formulate research hypotheses.

- Materials and method section is well described and correspond to the aim set out in the manuscript.  
- The tables and figures clearly presenting the obtained results with their appropriate interpretation. 
- Discussion is hard to understand please reframe the "Results and discussion" section.
- The references are sufficient and necessary. 

- The paper needs some editorial corrections.

- I recommend the publication of this manuscript in the Plants journal  after minor revisions.

Author Response

- In general terms the topic of the article is interesting, the methodology is explicitly presented and the results reported are interesting. 

Reply authors: We are very grateful for this positive comment

- In my opinion, the introduction chapter should end with a paragraph indicating the purposefulness of the conducted research. Authors should clearly define the purpose of the work and formulate research hypotheses.

Reply authors: We agree in general terms, although we decided to end the Introduction with information concerning some major results (lines 171-180) and, as a last sentence, the purposefulness of our study (lines 180-184), while the purpose (aims) is mentioned first as a very general aim in lines 99-100 (before justifying the selected model of study), and subsequently, the aims of our specific study design are defined in lines 147-153, after the compilation of previously published work.

- Materials and method section is well described and correspond to the aim set out in the manuscript.  
- The tables and figures clearly presenting the obtained results with their appropriate interpretation. 
- Discussion is hard to understand please reframe the "Results and discussion" section.
- The references are sufficient and necessary. 

We thank the referee for this positive evaluation although – after discussing the manuscript with several colleagues and students – we decided to maintain the combined ’results and discussion’ format rather than separating these two sections.

- The paper needs some editorial corrections.

We hope that we found and corrected all spelling errors and incomplete sentences.

Round 2

Reviewer 1 Report

Authors have sincerely addressed all the comments, but still the Ms is not as per the journal guidelines. Results and discussion should be separated.